# Proteolysis Degree of Protein Corona Affect Ultrasound-Induced Sublethal Effects on *Saccharomyces cerevisiae*: Transcriptomics Analysis and Adaptive Regulation of Membrane Homeostasis

**DOI:** 10.3390/foods11233883

**Published:** 2022-12-01

**Authors:** Zi-Yi Zheng, Chao-Hua Feng, Guo Xie, Wen-Li Liu, Xiao-Lei Zhu

**Affiliations:** School of Material Science and Food Engineering, University of Electronic Science and Technology of China, Zhongshan Institute, Zhongshan 528402, China

**Keywords:** protein corona, proteolysis, nanoparticle, ultrasound

## Abstract

Protein corona (PC) adsorbed on the surface of nanoparticles brings new research perspectives on the interaction between nanoparticles and fermentative microorganisms. Herein, the proteolysis of wheat PC adsorbed on a nano-Se surface using cell-free protease extract from *S. cerevisiae* was conducted. The proteolysis caused monotonic changes of ζ-potentials and surface hydrophobicity of PC. Notably, the innermost PC layer was difficult to be proteolyzed. Furthermore, when *S. cerevisiae* was stimulated by ultrasound + 0.1 mg/mL nano-Se@PC, the proportion of lethal and sublethal injured cells increased as a function of the proteolysis time of PC. The transcriptomics analysis revealed that 34 differentially expressed genes which varied monotonically were related to the plasma membrane, fatty acid metabolism, glycerolipid metabolism, etc. Significant declines in the membrane potential and proton motive force disruption of membrane were found with the prolonged proteolysis time; meanwhile, higher membrane permeability, membrane oxidative stress levels, membrane lipid fluidity, and micro-viscosity were triggered.

## 1. Introduction

Fermentation techniques using enzymatically active microbes have distinct advantages including low cost, reusability, strong adaptability, and mild reaction conditions in producing high-value compounds or degrading worthless components. Nevertheless, the low transport efficiency of the substrate into the microbial intracellular enzyme reaction centers is still a momentous restriction. Thus, approaches aiming to improve the permeability of cell membrane and simultaneously retain cell viability are studied, including permeabilizing solvents and detergents, intense submicrosecond electrical pulses, non-thermal atmospheric-pressure plasmas, micrometer-sized graphene oxide, low frequency ultrasound irradiation, etc. [1,2]. Among these, ultrasound (US)-guided shockwaves could alter the liquidity of the phospholipid bilayer, and might create nonlethal transient sonopores on cell membrane, thus facilitating cellular uptake [3]. 

Biocompatible nanoparticles combined with low intensities of US treatment were shown to provide satisfactory synergy effects in fermentation according to our previous work [4]. Nanoparticles that act as nucleation sites for ultrasound-triggered microbubbles could deposit directly on a cell surface (mechanism of sonoprinting) [5]. The microbubbles and the rigid surface of the near nanoparticles collide with the cell membrane, resulting in more intense dynamic effects of the stretch-compression scattering, reflection wave jet (water hammer), and wicket wave. 

In a fermentation liquor, nanoparticles with ultrahigh specific surface area easily adsorb the surrounding protein to form protein corona (PC), whose surface properties defines the real biological identity of nanoparticles [6]. Specifically and notably, the effects of extracellular proteases from fermenting bacteria on the PC surface properties and resultant synergy with US is a worthy of serious investigation. Protease might degrade the PC through the unique cleavage of peptide bonds, thus affecting the topological defect of the surface structure, viscoelasticity of nanoparticle@PC, and the surface hydrophobic properties (water contact angle). These changes might have a profound effect on the dynamic characteristics of microbubbles and the interface force at the protein/cytoderm or cytomembrane contact (electrostatic force, van der Waals force, hydrophobic force, etc.). 

Nano-Se is a better form of Se supplementation than inorganic salt or organic forms [7]. Recent studies exhibit the potential applications in natural defenses against cancer and inflammatory conditions of biogenic Se nanoparticles that are synthesized and stabilized by microbial or plant extracts [8,9]. In addition, Se is critical for the production of glutathione peroxidase, which might protect fermentative bacteria from oxidative damage of US [10].

Wheat proteins have classically been subdivided into albumin (water-soluble), globulin (salt-soluble), gliadin (alcohol-soluble), and glutenin (residual proteins) [11]. Notably, wheat glutenin, which accounts for around 45% of the total wheat protein, is a high hydrophobic heterogeneous mixture consisting of various subunits connected by disulfide bonds. Therefore, hydrophobic proteins, similar to conventional hydrophobic surfactants, might be excellent stabilizing agents for nanoparticles [12].

Herein, the response of *saccharomyces cerevisiae* including sublethal effects and cell morphology to US combined with nano-Se@PC–which was treated with time-limited proteolysis–was investigated at first. Then, the RNA-Seq analysis revealed mechanisms of molecular response and guided us to focus on the homeostasis regulation of the cell membrane. Therefore, the membrane permeability, membrane potential, proton motive force, oxidative stress levels in cell membranes, and membrane fluidity of *saccharomyces cerevisiae* cells were studied.

## 2. Materials and Methods

### 2.1. Materials

Raw wheat bran (WB) and wheat flour were purchased from Yihai Kerry Food Industry Co, LTD (Dongguan, China). Sodium selenite was purchased from Sinopharm Chemical Reagent Co., Ltd. (Shanghai, China). Ultra-pure water was produced by Milli-Q water purification system (Millipore, Milford, MA, USA). Bis-(1,3-dibutybarbituric acid) trimethine oxonol (Di-BAC_4_(3)), 3,3′-dipropylthiadicarbocyanine iodide (DISC_3_(5)) and BODIPY^581/591^ C_11_ were purchased from Thermo Fisher Scientific (Waltham, MA, USA). 1,6-diphenyl-1,3,5-hexatriene (DPH) was purchased from Sigma-Aldrich Co. (St Louis, MO, USA). 

### 2.2. WB Extract-mediated Preparation of Nano-Se

First, 10 g of dry WB was washed twice using tap water and soaked for 4 h with 200 mL of ultrapure water. After reflux for 4 h, the obtained WB extract was filtered twice and lyophilized. For the synthesis of nano-Se, 1 mL of Na_2_SeO_3_ solution (0.05 M), 10 mL of WB aqueous extract (8 mg/mL) were mixed. Then, 1 mL of ascorbic acid solution (0.15 M) was added dropwise into the mixture and then stirring for 10 h. The obtained solution containing nano-Se was dialysed (molecular weight cut off: 8–14 KD) in ultrapure water for 30 h to remove the excess Na_2_SeO_3_. The obtained nano-Se was lyophilized and stored at –4 °C until further use.

### 2.3. Extraction of Wheat Protein 

Wheat proteins were extracted from 1 g of flour by *Osborne* extraction procedure, with sequential extraction of albumins (distilled water), globulins (0.1 M NaCl), gliadins (70% ethanol), and glutenins (1.76 M NaOH). The quantification of protein fractions was conducted using a Pierce^TM^ BCA Protein Assay Kit (Thermo Fisher Scientific Inc., Waltham, MA, USA). The above four fractions were combined and lyophilized. To remove the salt, the lyophilized proteins were dialyzed with 3.5 kDa membranes (Thermo Fisher Scientific Inc., Waltham, MA, USA) in ultrapure water for 6 h at 10 °C, and the dialysis was repeat six times. 

Next, the obtained wheat proteins solution (5 mg/mL) was incubated with 0.3 mg/mL of prepared nano-Se at room temperature for 2 h under gentle agitation. The nano-Se along with the adsorbed protein corona (denifed as nano-Se@PC) were separated by centrifugation (15,000 rpm, 8 min, 4 °C) and washed twice using ultrapure water.

### 2.4. Preparation of Cell-free Protease Extract (CFPE)

The *S. cerevisiae* strain was seeded on YPD liquid medium and incubated for 12 h (32 °C, 150 rpm). Then, the *S. cerevisiae* (OD_600_ = 1.2) was harvested using centrifugation (4000 rpm, 10 min, 4 °C). The supernatant was filtered by 0.22 μm membrane and used as CFPE. 

Sigma’s non-specific protease activity assay was conducted using casein as a substrate. One unit (U/mg) of protease activity of CFPE was defined as the amount of enzyme capable of releasing 1 μmol of tyrosine per mL per minute [13]. 

### 2.5. CFPE Proteolysis of Nano-Se@PC

Nano-Se@PC (Se concentration of 0.2 mg/mL) were incubated in 5 mL of CFPE (protease activity of 235.8 u/mL) under gentle stirring at 32 °C and the maximum proteolysis time was set to 12 h. The residual proteins on the surface of nano-Se were fully dissociated by a loading buffer (2 mmol/L DTT, 50 mmol/L Tris-HCl, 0.15 mol/L SDS) combined with the assistance of US, then the proteolysis degree of PC was quantified using Pierce^TM^ BCA Protein Assay Kit (Thermo Fisher Scientific Inc., Waltham, MA, USA). The mean hydrodynamic diameters and the ζ-potentials were measured by dynamic light scattering (DLS), and the morphology were obtained by transmission electron microscope (TEM).

### 2.6. Measurement of PC Degree of Hydrolysis and Surface Hydrophobicity 

The OPA method with some modifications was used to determine the degree of hydrolysis (DH) of protein corona. First, 1 mL of 0.1 M sodium tetraborate decahydrate solution containing 0.02 mM SDS, 2 μL of β-mercaptoethanol, 20 of methanol-OPA (1:250, *w*/*v*), and 50 μL of proteolytic protein corona solution (0–12 h) or cell-free protease extract from *S. cerevisiae* were mixed. After 2 min, absorbance at 340 nm was measured, and glycine was used as standard. DH values were calculated as the following formula:$$DH\%=\frac{NH_{2ti}-NH_{2t0}-NH_{2ti.CFPE}}{NH_{2Total}-NH_{2ti.CFPE}}\times 100$$

NH_2ti_ and NH_2t0_ were the free amino groups at i and 0 h. NH_2ti.CFPE_ was the free amino groups in CFPE solution at i h. NH_2Total_ was the free amino groups from the whole protein corona.

To avoid the interference of electrostatic or π–π interaction between organic dye and PC, octanol–water partition coefficients (K_OW_) of nano-Se@PC were adopted to characterize the alteration of surface hydrophobicity. Next, 5 mL of Octanol and 5 mL of water were mixed, and equilibration process started with the addition of nano-Se@PC (0.5 mg). After 20 h, the liquid mixture was shaken at 50 rpm for 4 h, followed static separation for 4 h. Nano-Se@PC was collected from each phase, and the concentration of nano-Se in each liquid mixture were quantified by Inductively Coupled Plasma-optical emission spectroscopy (ICP-OES, Agilent Technologies Inc., Santa Clara, CA, USA). 

### 2.7. Co-treatment of US and Nano-Se@PC on S. cerevisiae

First, 3 log CFU/ml of *S. cerevisiae* and CFPE-treated nano-Se@PC (Se concentration of 0.1 mg/mL) were treated by 45 kHz ultrasonic waves from an ultrasonic bath. The amplitude was 50% (0.0461 W/mL) and action period was 15 s under continuous mode. The parameter values of US and nano-Se@PC were chosen according to previous experiments about sublethal concentration (LC_10_) of *S. cerevisiae*, in order to insure the further transcriptome sequencing. The power of US was calculated by the calorimetric method as per the following equation:$$Power=\frac{dT}{dt}MC$$
where *m* and *C* are the molar mass and heat capacity of water at 30 °C, *dT*/*dt* is the slope of the mixture temperature versus the ultrasound time [14]. 

### 2.8. Evaluation of Lethal and Sublethal Injury

The number of sublethal injured *S. cerevisiae* were estimated as the difference in the numbers of CFU between US + nano-Se@PC-treated *S. cerevisiae* cells on the nonselective (YPD agar) and selective (YPD + 7% of NaCl) media. Overall, 7% of NaCl was the highest level which did not significantly affect the *S. cerevisiae* count under normal growth conditions in comparison with nonselective media. The lethal injury of the treatment was estimated as the difference in the number between the untreated and treated cells which recovered on the YPD agar.

### 2.9. RNA Extraction, Sequencing and Transcriptomics Analysis

After US + nano-se@PC treatments, *S. cerevisiae* were separated by centrifugation (5000 rpm, 4 min). The total RNA of *S. cerevisiae* was extracted using RNeasy Mini Kit (Qiagen, Germantown, MD, USA) according to the manufacturer’s protocol, and treated with RNase-free DNase Ⅰ (Takara) to remove contaminating genomic DNA. Then, RNAs were quantified at 260 nm, and the integrity of RNAs was evaluated by formaldehyde agarose gel electrophoresis. RNA-Seq profiling were conducted by BGI (Beijing Genomics Institute, Shenzhen, China). Briefly, the mRNA was separated from the total RNA and short fragments were obtained by adding fragmentation buffer. The cDNA was obtained by reverse transcription using random hexamer-primers, followed by the synthesis of the second cDNA strand using DNA polymerase Ⅰ and RNase H. After purification and amplification, the cDNA library was created and sequenced, respectively, on the Illumina Hi-Seq 2000 platform. The raw data were filtered via deleting the adaptor and low-quality sequences. The differentially expressed genes (DEGs) were defined based on the false discovery rate (FDR) < 0.05 and log2 (fold change) ≥ 1. The enrichment analysis of DEGs were performed based on Gene Ontology (GO) and Kyoto Encyclopedia of Genes and Genomes (KEGG), and Bonferroni-corrected *p* < 0.05 was considered statistically significant. 

### 2.10. Quantitative Validation by Real-Time PCR (qRT-PCR)

Overall, 7 DEGs were selected randomly for qRT-PCR analysis to confirm the results of RNA-Seq. Extracted RNA was reverse transcribed into cDNA using PrimeScript^TM^ RT-PCR Kit (Takara, Japan). Primers of target and reference genes were listed in Table 1. Three biological replicated were conducted.

### 2.11. Measurement of Cell Membrane Integrity

First, 1 mL of *S. cerevisiae* cells was stained by 1 μL of propidium iodide (1 mg/mL) in the dark. After 10 min, the abovementioned treatment of US + nano-Se@PC was conducted. Then, the cells were centrifuged, washed, and resuspended in PBS to remove excess PI. The cellular uptake of PI was analyzed using FACSCaliber flow cytometer (BD Biosciences). The excitation and emission wavelengths were 488 nm and 620 nm, respectively. The same volume of ultrapure water and 1% Triton X-100 were used as the blank control and the positive control, respectively.

### 2.12. Measurement of the Membrane Potential

The treated cells (OD_600_ = 0.5) were stained by 2 μg/mL Di-BAC_4_(3) in black and non-transparent 96-well plates at 35 °C for 40 min. The cell membrane potential was detected using flow cytometry (excitation wavelength of 488 nm, emission wavelength of 530 nm).

### 2.13. Proton Motive Force Assay

The treated cells (OD_600_ = 0.2) were incubated with 0.3 μM DISC_3_(5) for 10 min. Measurements were conducted in black and non-transparent microtiter plates using a fluorimeter (622 nm excitation and 670 nm emission filters).

### 2.14. Oxidation Kinetics of Membrane Lipid 

First, 1 mM of the lipid oxidation-sensitive probe-BODIPY^581/591^ C_11_ was dissolved in DMSO. *S. cerevisiae* cells were suspended in 10 mM citrate buffer (Ph = 7) and 10μM C_11_-BODIPY^581/591^ solution was added. After incubation for 30 min at 35 °C and 120 rpm in the dark, the cells were treated by nano-Se@PC + US. Then 0.2 mL of lysozyme (5 mg/mL) and 0.4 mL EDTA (0.2 M) were added to the suspension in order to increase the membrane-solubility of the probe. The sample that had an equal amount of citrate buffer added instead of nano-Se@PC + US was as the blank control. The membrane lipid peroxidation kinetics were monitored using fluorescence spectrophotometer at 500 nm (excitation wavelength) and 520 nm (emission wavelength). 

### 2.15. Measurement of Membrane Fluidity

The variation of membrane fluidity was assessed using DPH as the fluorescence probe [15]. First, 2 μM of DPH was dissolved in tetrahydrofuran and the mixture was dissolved in PBS. The treated cells were obtained by centrifugation (5000 rpm, 8 min) and rinsing twice in PBS. Subsequently, the cell pellets were incubated in the prepared DPH solution at 35 °C for 40 min. Excessive DPH were removed by centrifugation (9000 rpm, 15 min, 4 °C). The obtained cells were washed and resuspended with PBS. Then, the cells were transferred to a black 96-well plate and measured by a multi-mode microplate reader (Perkin Elmer, Waltham, MA, USA, Ex/Em = 362/432 nm). Fluorescence polarization (*P*), membrane micro-viscosity (*η*), and membrane lipid fluidity (MLF) were calculated based on the following equations:$$P=\frac{I_{vv}-GI_{vh}}{I_{vv}+GI_{vh}}$$
$$\eta =\frac{2P}{0.46-P}$$
$$MLF=\frac{0.5-P}{P^{2}}$$
where I*vv* and I*vh* are the light intensities emitted in the vertical and horizontal directions relative to the beam of excitation. *G* is the grating factor.

## 3. Results and Discussion

### 3.1. Characterization of Nano-Se and Nano-Se@PC

During the WB extract-mediated preparation process of nano-Se, it was observed that the color of the liquid mixture gradually changed from colorless to ruby red in color, indicating that WB extract containing phenolics and flavonoids, which act as reducing agents, could reduce sodium selenite to the elemental form of Se. An FTIR analysis was conducted to confirm the reductive groups in WB extract. As shown in Figure 1a, the FTIR spectrum of WB extract demonstrated that characteristic peaks at 2972 cm^−1^ was specified for -CH_3_ groups in ferulic acid [16]. The peak at 3389 cm^−1^ was the stretching vibration of bonded hydroxyl group of phenols [17]. In the FTIR spectrum of the prepared nano-Se, two new bands at 1635 cm^−1^ and 1228 cm^−1^, which corresponded to amide Ⅰ and amide Ⅲ groups, demonstrated the presence of wheat proteins. 

X-ray energy dispersive spectroscopy (EDS) results further confirmed the presence of Se, C, and O elements (Figure 1b, Appendix A). The pattern of XRD was shown in Figure 1c. The peaks at 23.5°, 30°, 41.2°, 43.8°, 45.5°, 51.8°, 61.6°, and 65° indicated the presence of crystalline Se in the nanoparticle structure (JCPDS card No. 06-0362).

The (TEM) images (Appendix A) showed that the typical prepared nano-Se was oval in shape, and DLS results showed that the nano-Se had an average diameter of 62.1 ± 16.7 nm (Figure 1d, green line). After incubation with wheat protein, the diameter of nano-Se@PC was up to 236 ± 30.2 nm (Figure 1d, black line). With the extension of proteolysis time (0–12 h), the diameter of nano-Se@PC gradually dwindled to 79.4 ± 12.9 nm and all nano-Se@PC remained in monodispersity. Moreover, the results of the mean surface charge density (ζ-potential) measurement showed that the ζ-potential of nano-Se@PC shifted towards lower values (from −3.6 mV to −7.7 mV) during proteolysis (Figure 1e). The enhancement of the surface charge density might improve the stability of nano-Se@PC and prevent aggregation in the fermentation liquor. Similarly, the surface hydrophobicity of nano-Se@PC varied monotonically with the proteolysis time (Figure 1f, blue line). K_OW_ (the octanol-water partitioning coefficient) value > 1 indicated a preference of surface hydrophobic. The higher K_OW_ values of nano-Se@PC_12_ and nano-Se@PC_2_ than that of nano-Se@PC_0_ explained the increased hydrophobicity during the proteolysis of PC, indicating that the proteolysis of CFPE might expose the hydrophobic regions of wheat protein which absorbed on the surface PC. Another possibility is that the hydrophobicity of the inner protein was higher than that of the protein in the outer layer of PC. 

In the aspect of proteolysis kinetics of PC, it was found that the amount of PC reduced from 8.61 ± 0.92 mg/mg (on the Se-weight basis) to 0.52 ± 0.07 mg/mg during 12 h of CFPE-proteolysis (Figure 1f, green line). Surprisingly, a slight increase in the PC amount within 2 h of proteolysis was found. This might stem from the fact that proteolysis in the early stage promoted the adsorption or incremental exchange of proteins (peptides) in the adjacent microenvironment onto the PC surface. Furthermore, the PC amount was not changed significantly after 7 h of proteolysis, indicating the presence of a non-proteolysis PC fraction on the contact surface with nano-Se. The steric hindrance and polymer shielding of the innermost layer protein might result in higher resistance to proteolysis.

Moreover, DH values of PC started with 0% and gradually increased to 91.6 ± 3.8% during 12 h of CFPE-proteolysis, as shown in the following figure (Figure 1f, red line).

### 3.2. Assessment of Lethal and Sublethal Effects of Nano-Se@PC+ US

The sublethal injury of a microorganism is usually described as any damage except death. Sublethal-injured *S. cerevisiae* experienced the transient loss of cell functions and was unable to grow on the selective medium, but could recover under favorite conditions [18]. According to the experiment results, both lethal and sublethal injury of *S. cerevisiae* occurred after treatments of US + nano-Se@PC_0_, nano-Se@PC_2_, or nano-Se@PC_12_. In comparison with US+nano-Se@PC_0_ at 0 h, subjecting the *S. cerevisiae* cells to US + nano-Se@PC_2_ and nano-Se@PC_12_ resulted in a reduction of 0.011 log CFU/ml and a significant decrease by 0.024 log CFU/ml in the numbers of the lethal-injured cells, respectively (Figure 2a). On the other hand, sublethal-injured cells number in PC_0_ group was significantly less than the numbers in PC_2_ and PC_12_ groups. Figure 2b showed the growth curves modelled by the modified Gompertz equation. The most extended lag-phase (*λ*) and the maximum specific growth rate (*μm*) were both found in the nano-Se@PC_12_ group. Nevertheless, it was found that the maximum bacteria concentrations in the stationary phase (*A*) were not significantly influenced by the different PC. 

Cell morphology was assessed using SEM. As shown in Figure 2c–f, the sizes of cells did not change with the alteration of the treatment. Among US, US+ nano-Se@PC_2_, and US+ nano-Se@PC_12_ groups, wrinkled, uneven, and lysed cells were clearly observed, and the maximum destructive was found in the US+ nano-Se@PC_12_ group. With the aid of certain nanoparticles which act as sonosensitizers on the surface of cell membrane, the acoustic cavitation induced by US improved the generation of ·OH, and ·OH could trigger integrity loss in the cell wall or membrane [19]. Surprisingly, it was observed that *S. cerevisiae* cells in US+ nano-Se@PC_0_ group had a more intact morphology and smoother surface compared with the other three groups (Figure 2d). This attenuated acoustic cavitation effect might be related to the loosely coupled architecture in the outermost layer of the PC, which was called “soft” protein corona.

### 3.3. Investigation of Cell Membrane Homeostasis 

The process of air-containing microbubbles oscillation and destruction under ultrasound produced microstreaming, microjects, and shock waves, which has been shown to improve the cytomembrane permeability [20]. However, the effect of nanoparticle@PC which interferes US-induced changes of cell membrane homeostasis is still unclear. The adaptive adjustment of cell membrane properties is vital for cell survival in complicated and variable microenvironments. Therefore, to further investigate the physiological states of live-cell membranes, PI-based fluorescence assay was applied to evaluate changes of the membrane permeability. It was found that US+ nano-Se@PC_12_ enhanced the cell membrane permeability level by 104% and 61% in comparison with those of nano-Se@PC_0_ and nano-Se@PC_2_ groups (Figure 3a). These results were also evidenced by the quantitative analysis of PI fluorescence using flow cytometry, which showed that the percentages of PI-positive cells were 39.1%, 26.9%, and 15.8% in the samples treated with US+ nano-Se@PC_12_, US+ nano-Se@PC_2_ and US+ nano-Se@PC_0_, respectively (Figure 3b). Based on the results of cell counting and membrane permeability experiments, it could be concluded that US-induced sublethal membrane damage was waned with the presence of non-proteolysis PC on the surface of nanoparticles, and the non-proteolysis PC might result in smaller membrane pores than full-proteolysis PC. Previous research showed that the small hole (a pore size < 50 nm) on the membrane could be repaired immediately in 5 to 120 s [21].

The effect of US+ nano-Se@PC on the membrane potential of *S. cerevisiae* was studied by measuring the fluorescence intensity of Di-BAC_4_(3). As shown in Figure 3c–d, it was observed that the maximum fluorescence intensity values decreased in a proteolytic time-dependent manner. In contrast with the results of the comparison between nano-Se@PC_0_ and nano-Se@PC_12_, the difference between nano-Se@PC_0_ and nano-Se@PC_2_ groups was not significant (*p* > 0.05). According to a previous study, it was supposed that in comparison with nano-Se@PC_0_ and nano-Se@PC_2_ groups, more generated free radicals in situ by nano-Se@PC_12_ +US might increase the peroxidation of cell membrane lipids, thus resulting in this diminished potential of cell membrane. 

As an additional experiment of the abovementioned experiment, lipophilic oxidation-sensitive C_11_-BODIPY^581/591^ was used as the valid probe to monitor the membrane lipid peroxidation. Figure 3e represented lipid oxidation kinetics of *S. cerevisiae* cytoplasmic membrane after treatments with different nano-Se@PC+ US. The highest ratio of fluorescence intensity to cell number was found in PC_12_ group at each time point. After 10 h, 15 h, and 25 h, the levels of membrane lipid peroxidation declined to the minimum values (10.02, 10.13, 10.68) in PC_0_, PC_2_, and PC_12_ groups, respectively. 

The proton motive force (PMF), which is equal to the sum of transmembrane electrical potential (Δ*ψ*) and transmembrane proton gradient (ΔpH), drives multiple vital pathways in cells [22]. It was envisaged that the dissipation of PMF, which was related with electron transport across the respiratory chain and ATP synthesis, might enhance cell penetration and the reduced efflux of metal-ion. Moreover, the dissipation of ΔpH or Δ*ψ* resulted in the collapse of PMF [19]. The disrupted Δ*ψ* will cause an increase in DISC_3_(5) fluorescence, and the opposite happens when ΔpH is disrupted. As shown in Figure 3f, all treatments led to a sharp rise in fluorescence value, signaling the dissipation of Δ*ψ*. In particular, the interactions between nanoparticles@PC_12_ and membrane under ultrasound were more remarkable than the other two groups. 

Membrane polarity and fluidity are different depending on the surrounding environment. The value of fluorescence polarization (P) is inversely proportional to the membrane fluidity [23]. Figure 3g showed that all treatment resulted in increased *p* values in a time-dependent and degree-of-proteolysis-dependent manner. In this regard, significant differences of the highest P values were found between the PC_12_ group and the other groups after 5 h, demonstrating that nano-Se@PC_12_ led to the biggest disruption of membrane fluidity. Furthermore, the membrane micro-viscosity (*η*), which was positively correlated with the membrane rigidity, was also calculated. As observed in Figure 3h, as the degrees of PC proteolysis increased, the values of *η* rose, also indicating the highest rigidity of the membrane in the PC_12_ group. Furthermore, compared with the PC_2_ group, nano-Se@PC_12_ +US treatment led to a decline in MLF (data not shown).

### 3.4. Transcriptional Response of S. cerevisiae to US + nano-Se@PC

From the above results, the surface charge density, rigidity, hydrophobicity of protein corona, and diameter of nanoparticle@PC might influence the membrane physicochemical homeostasis of *S. cerevisiae* under ultrasound, and full proteolytic PC–which might provide better mutual lipophicity or charge complementarity to the cell membrane–led to the most intense alterations of cell membrane properties. 

Ulteriorly, we applied transcriptome analysis to reveal “invisible” alterations at the genetic level in *S. cerevisiae* exposed to U + nano-Se@PC. Using U + nano-Se@PC_0_ as a reference, 296 DEGs (162 upregulated and 134 downregulated) were identified in PC_2_ group (| abs(log2 (fold-change)) ≥1, *p* ≤ 0.05 and FDR ≤ 0.01, Figure 4a), and 430 DEGs (247 upregulated and 183 downregulated) were identified in PC_12_ group (Appendix A). In the third pairwise comparison (PC_2_ vs. PC_12_), 192 DEGs (105 upregulated and 87 downregulated) were identified (Appendix A). PCA analysis was conducted and reported in Figure 4b. PCA analysis covered nearly 76% of the variability in all samples and the samples corresponding to replicate measurements were found to be grouped together. It was observed that there was an overlap between PC_0_ and PC_2_ groups and a complete separation between PC_12_ and PC_0_ groups.

To facilitate the understanding of the correlation between the regulatory network of genes and changes in the diameter of nano-Se@PC (D), ζ-potential (Z), surface hydrophobic (Kow), the content of PC (C), number of sublethal-injured *S. cerevisiae* (NS), number of lethal-injured *S. cerevisiae* (NL), membrane permeability (MP), membrane potential (MZ), membrane lipid peroxidation (MLP), proton motive force (PMF), membrane fluidity (MF) and membrane rigidity (MR), a weighted gene co-expression network analysis (WGCNA) was conducted. A total of 17 co-expression modules which were marked with different colors were subsequently identified, with the gene number in each module ranging from 324 to 15 (Figure 5a). As shown in Figure 5b, the turquoise module (34 genes) was negatively correlated with membrane permeability (r = –0.29, *p* = 0.0006), while the black module (67 genes) had a positive relationship with ζ-potentials of PC (r = 0.31, *p* = 0.0003). The changes in Kow were most correlated with gene expression in the grey, yellow, and black modules (*r* > 0.43, *p* < 6 × 10^−6^). Gene expression levels in the yellow module had a low presence of low surface hydrophobic of PC, and then gradually ascended with the increase in surface hydrophobic of PC. In contrast, genes expression levels in the green modules showed the opposite pattern. Notably, the gene expression in the pink, red, cyan, and salmon modules were almost impervious to the properties of PC. We also found no modules were associated with the number of sublethal-injured *S. cerevisiae*, number of lethal-injured *S. cerevisiae*, membrane lipid peroxidation, and proton motive force, implying that the regulation of some cell membrane characteristic and survival status is not likely to be simply a matter of gene co-expression module but of a few specific genes. According to the limited research, the modulatory effects of nanoparticle size or surface charge were essential to nanoparticle–cell interactions, especially toxicity. On this basis, our findings suggested that the surface hydrophobicity of nanoparticle@PC might exhibit a more important role on the gene expression of cell than other properties of nanoparticle@PC, including diameter, ζ-potential, and the content of PC. 

After analysis of the regulatory gene network of *S. cerevisiae* response to different PC, KEGG pathway and GO analysis were conducted to facilitate the understanding the transcriptional regulatory function of the identified DEGs. First, the 20 KEGG pathway entries with the most significant enrichment in three pairwise comparisons were shown in Figure 4c and Appendix A, respectively. It was found that the significantly enriched pathways of DEGs in the first pairwise comparison (PC_2_ *vs.* PC_0_) were metabolic pathways, glycolysis/gluconeogenesis, endocytosis, fatty acid metabolism, glycerolipid metabolism, etc. Additionally, significantly enriched pathways of DEG in the comparison of PC_12_ *vs.* PC_0_ were lipid metabolic, fatty acid biosynthesis, etc. Moreover, KEGG analysis indicated that 21 genes in the yellow module were enriched in the fatty acid metabolism pathway, and seven genes in the black module were enriched in the glycerolipid metabolism pathway. 

Furthermore, chord diagrams for GO enrichment analysis showed 10 most enriched GO terms (at least six genes for each term). In comparison, in the PC_0_ with PC_2_ group, 29 significantly (*p* < 0.01) enriched biological processes were found, especially the small molecule metabolic process (GO: 0019220), oxidation reduction (GO: 0006468), small molecule biosynthetic process (GO: 0009165), cellular lipid metabolic process (GO: 0007242), cellular metabolic process (GO: 0009199), and response to oxidative stress (GO:0006796). Overall, 11 enriched cellular components and three molecular functions were also found in this pairwise comparison. In comparison, in the PC_0_ group and PC_12_ group, the analysis of biological process ontology revealed 36 significantly enriched terms, including the lipid metabolic process (GO: 0007242), lipid biosynthetic process (GO:0001568), response to oxidative stress (GO:0048514), phospholipid metabolic process (GO:0006796), etc. Significantly enriched cellular component ontologies were cell redox homeostasis (GO:0005886), transferase activity, transferring acyl groups (GO:0005912), plasma membrane enriched fraction (GO:0044459), etc. For molecular function ontologies, the top three significantly enriched functions were the phospholipid metabolic process (GO:0060589), nucleotide biosynthetic process (GO:0030695), and nucleotide biosynthetic process (GO:0030695). Based on the combined analyzing between WGCNA, KEGG, and GO, fine-tuned properties and functions of membranes in response to different PC were probably associated with the membrane lipid.

On the other hand, 34 genes which were monotonously upregulated or downregulated with the increase in proteolysis time were analyzed. For instance, the two highest upregulations (log2 fold-change = 3.16 and 3.09, respectively) among these 34 DEGs were *OLE1* and *PDR16*. *OLE1* (acyl-CoA desaturase involved in desaturation of fatty acid) was annotated to ‘lipid metabolism’, and it was reported that membrane fluidity was variably regulated by *OLE1* [24]. The upregulation of *PDR16* which involved in phospholipid synthesis might increase cell surface hydrophobicity [25]. *YPT53* (2.2-fold) and *LAS17* (1.7-fold), which were in the tan module, have been shown to regulate the organization of cytoplasmic structures [26]. *YPT53* was also required for the endocytic of yeast, indicating that PC with different proteolysis level might also affect the *S. cerevisiae* endocytosis of nanoparticles [27]. Similarly, *LAS17* is the primary activator of Arp2/3-driven actin nucleation, which is demanded in membrane invagination during endocytosis [28]. The expression of *SSA1* and *SSA4* which were involved in membrane and cytoplasmic proteins were both in the blue module, and upregulated in the PC_2_ and PC_12_ group [29]. The expression levels of *ETR1* and *SUR4* which showed a declining trend with the proteolysis time of PC were related to saturated fatty acid metabolism [30]. *CDS1*, *AYR1*, and *PDX3* (black module) which were involved in the transcriptional regulation of lipid and phospholipid metabolism, were found to be monotonously downregulated. In the same module, *FEN2*, *PEX15*, and *URA8* (cytidine triphosphate synthetase) were involved in the regulation of isoprenoid metabolism and the synthesis of membrane phospholipids [31]. These six genes (*CDS1*, *AYR1*, *PDX3*, *FEN2*, *PEX15*, and *URA8*) were among the top 10 hub genes (ranked by connection degree) in the black module (data not shown). Ergosterol and sterol biosynthesis genes (*ERG1, ERG12, HMG1,* and *MVD1*) were also found to be upregulated with the increase in the proteolysis time [32]. 

Only one gene (*HOR7*) related to the cell wall was found to be monotonously downregulated [33]. Heat Shock Proteins (*HSP30, HSP42*), which have been implicated in membrane properties, were found to be upregulated [34]. On the other hand, antioxidant response-related genes including *GTT1* (glutathione-S-transferases), *ECM4* (xi-class glutathione transferase), *GPX1* (glutathione peroxidase), *RHR2* (glycerol-1-phosphatase), *MXR1, GPX2, NCE103,* and *LTV1* showed the same change tendency with the increasing proteolysis time of PC [35,36]. Combining the results of membrane lipid peroxidation and the abovementioned transcriptome analysis, it could be inferred that the proteolysis of PC might affect the ROS level produced by ultrasound and the consequent oxidative stress of *S. cerevisiae*. According to a previous study, the upregulation of saturated fatty acids could protect the cell against ROS [37]. Collectively, these results emphasized the disparate effect of PC in the US-induced self-adaptive regulation of cell membrane homeostasis. In addition, it was hypothesized that ultrasound treatment might allow stronger interactions between nanoparticles@PC with higher surface hydrophobicity and *S. cerevisiae* membranes, substantially enhancing its inherent biocidal effect and ultimately leading to more cell death. 

To further confirm the DEGs detected by RNA-Seq, the alterations of the expression levels of seven randomly selected DEGs were examined by qRT-PCR. Although the relative expression levels of the selected genes were slightly different, the results between RNA-Seq and qRT-PCR present good correlation (*R*^2^ = 0.8969, Table 1).

## 4. Conclusions

The present work recapitulated the considerable influences of protein corona absorbed on the surface of nanoparticles, which acted as the acoustic cavitation regulator in ultrasound-affected fermentation process. Cooperating with full proteolytic PC, ultrasound was shown to cause more numbers of the sublethal-injured compared with PC of a lower proteolysis degree. The level of discontinuities on the cell walls, integrity of the cell membrane, and misshape of cell structures induced by ultrasound was highly dependent on the proteolysis degree of PC. Furthermore, the results of transcriptomic analysis and weighted gene co-expression network analysis showed that the Kow value and ζ-potential of nano-Se@PC impacted four co-expression modules, including 206 genes, which were mainly enriched in fatty acid metabolism and glycerolipid metabolism pathways. In addition to this, our observations regarding cell membrane properties aided in interpreting the underlying mechanisms of correlation between gene expression patterns and membrane permeability, membrane potential, membrane lipid peroxidation, proton motive force, membrane fluidity, and membrane rigidity. 

## Figures and Tables

**Figure 1 foods-11-03883-f001:**
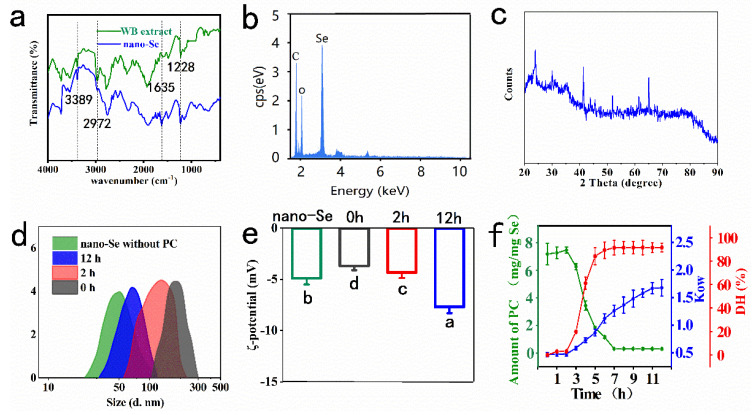
(**a**) FTIR spectra, (**b**) EDS spectra, and (**c**) XRD spectra of the prepared nano-Se. (**d**) Changes of average hydrodynamic diameters and (**e**) ζ-potential of during PC proteolysis using CFPE. (**f**) Changes in the PC amount on the surface of nano-Se and surface hydrophobicity with the increase in proteolysis time. Error bars represent the standard deviation (*n* = 6). Different alphabetic letters indicate significant difference (*p* < 0.05).

**Figure 2 foods-11-03883-f002:**
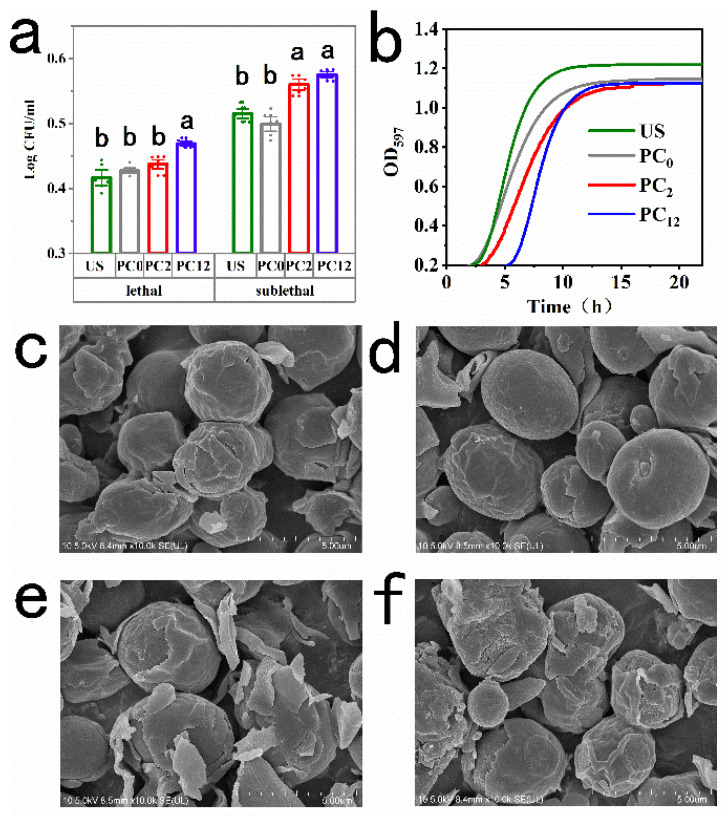
(**a**) Numbers of lethal and sublethal injuried *S. cerevisiae* when exposed to US, nano-Se@PC_0_, nano-Se@PC_2_, or nano-Se@PC_12_ combined with US at 0 h, (**b**) the corresponding growth curves and (**c**–**f**) SEM viewing of *S. cerevisiae* cells. Error bars represent standard deviation (*n* = 6). Different alphabetic letters indicate significant difference (*p* < 0.05).

**Figure 3 foods-11-03883-f003:**
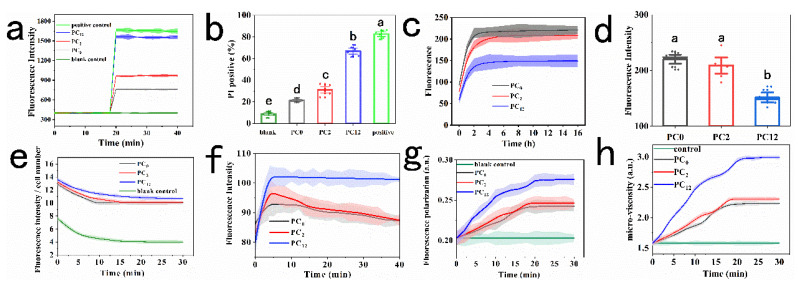
(**a**) Dynamic fluorescence of PI-staining *S. cerevisiae* cells following the treatments of nano-Se@PC_0_, nano-Se@PC_2_, or nano-Se@PC_12_ combined with ultrasound. (**b**) *S. cerevisiae* cells membrane permeability at 20 min. (**c**) Dynamic fluorescence of DISC_3_(5)-treated *S. cerevisiae* cells. (**d**) Membrane potential at 10 min. (**e**) Dynamic changes in fluorescence intensity of BODIPY^581/591^ C_11_)/cell numbers. (**f**) Dynamic changes of Δ*ψ*, (**g**) dynamic changes of fluorescence polarization, and (**h**) membrane micro-viscosity of *S. cerevisiae* cells treated with different nano-Se@PC + ultrasound. Different alphabetic letters indicate significant difference (*p* < 0.05).

**Figure 4 foods-11-03883-f004:**
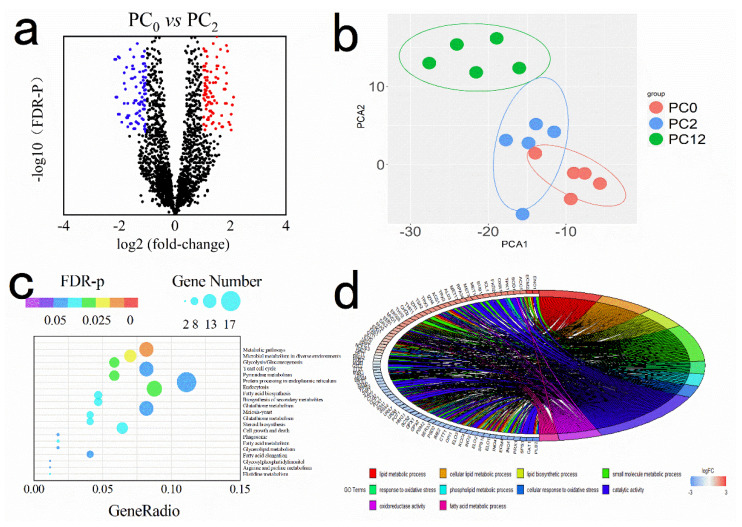
(**a**) Volcano plots showing the distribution of DEGs between U+nano-Se@PC_0_ and U+nano-Se@PC_2_, arranged by fold-change (x axis) and FDR-p values (y axis). DEGs are colored blue (down-regulation) and red (up-regulation). (**b**) PCA of normalized value of all genes in PC_0_, PC_2_ and PC_12_ groups (5 replicates in each group). (**c**) KEGG pathway enrichment analysis of DEGs between U+nano-Se@PC_0_ and U+nano-Se@PC_2_. (**d**) GO classification of DEGs between PC_0_ and PC_2_ groups.

**Figure 5 foods-11-03883-f005:**
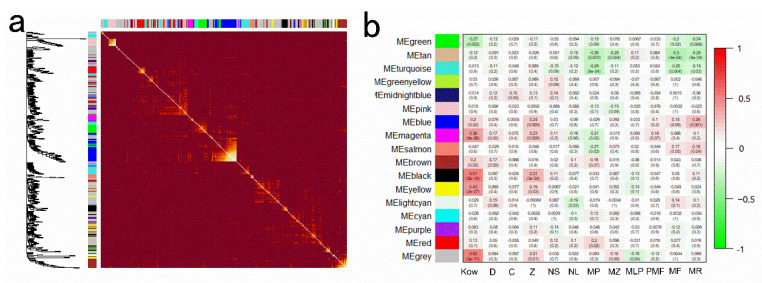
Weighted gene co-expression network analysis. (**a**) Cluster dendrogram of gene; each line represents one gene. (**b**) Module–trait relationships. The correlation coefficients and P values were exhibited in each cell.

**Table 1 foods-11-03883-t001:** Primers for qRT-PCR and verification of RNA-Seq results.

Gene Name	Forward Primer (5′-3′)	Reverse Primer (5′-3′)	qRT-PCR Results (log2FC)	RNA-Seq Results (log2FC)
*MKS1*	TTTTAACTCGGCCAATGACATCACC	AATTGTCTGTTTGGAGCAACGTCAT	4.79	3.01
*YBR204C*	AAATGGTCTACAGCGGTGCC	TGACATGCCAGAAAACAACCC	4.8	5.98
*CYS4*	TCGACTTAGTTGGTAACACCCCATT	TGGCAATTCTGTCTTTGATGGAACC	0.24	0.37
*YCR06W*	GATACGAGCAGGCTGCCAAG	GAGCCTCGATGAGGATTCCC	0.16	0.31
*FAS1*	AATTCAAAGCCACCCACATA	AGTACCGGCAACGATAACAC	0.22	0.39
*GPX2-R*	GCTTGGGTTGCTGTTGTTTC	ACAGGCTTTGGATTTCTTGG	5.41	4.12
*YDL10C*	CCCGACGCAAAGTTCTTCTC	AGTGTTGGTGATGGAGGCG	4.41	2.89

## Data Availability

The data presented in this study are available within the article and Appendix A.

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
