# Peer review of "Proteolysis Degree of Protein Corona Affect Ultrasound-Induced Sublethal Effects on Saccharomyces cerevisiae: Transcriptomics Analysis and Adaptive Regulation of Membrane Homeostasis"

_foods, 2022, doi:10.3390/foods11233883_

Round 1
Reviewer 1 Report
The presented article is devoted to the proteolysis by cells of wheat proteins located on the surface of nanoparticles (protein corona) with the intensification of the process by ultrasonication. The specificity of the study lies in the complexity of the phenomenon under consideration, in which all components of the system change during proteolysis, namely, wheat proteins are hydrolyzed, their fragments are desorbed from the surface of nanoparticles, the properties of cell membranes change, both under the action of ultrasound and under the action of proteolysis. Proteolysis by one proteolytic enzyme of one well-purified protein is a rather complex phenomenon in itself. The authors used a mixture of proteins that are hydrolyzed by cells, i.e., a mixture of proteases, which noticeably complicates the study. Therefore, details are not always given, for example, regarding the products of proteolysis and the kinetics of the process.
I think that the article can be improved taking into account the following comments and suggestions:
- The introduction is mainly devoted to the physicochemical features of the fermentative microorganisms in the presence of protein corona. It is necessary to show what relation this phenomenon has for Food Science. Can we expect that the proposed method of proteolysis can have advantages?
- In the introduction, the “low transport efficiency of the substrate into the microbial intracellular enzyme reaction centers” is mentioned. But nothing is said about methods that facilitate the release of proteases through cell membranes.
- The title of the article refers to the “degree of proteolysis”. But in fact, this characteristic is not defined by the authors. Only Figure 1f presents data on the reduction of protein on the surface of nanoparticles. In general, it is customary to use the degree of hydrolysis (DH) as a quantitative characteristic of the progress of proteolysis. The data presented in Figure 1f include information on both processes of hydrolysis of peptide bonds and desorption of peptide fragments.
- Lines 236, 245. Should be Figure 1f instead of Figure 1g.
Author Response
- The introduction is mainly devoted to the physicochemical features of the fermentative microorganisms in the presence of protein corona. It is necessary to show what relation this phenomenon has for Food Science. Can we expect that the proposed method of proteolysis can have advantages?
ANSWER: Thank you for your suggestions. Nanoparticles (NPs) such as food grade SiO2, TiO2 are copiously incorporated into different food products. Protein corona (PC) forms upon NPs as they enter the fermentation broth or human mouth. Significant differences in corona composition, and protein abundance has been shown to impact amylase and lysozyme activities of saliva, antigenicity and allergenicity of milk proteins, adherence of NPs to intestinal epithelial cells [1, 2].
Proteolysis of PC changes corona composition, protein abundance, surface properties, which define the real biological identity of nanoparticles. These alterations have an enormous influence on the gastrointestinal fate and potential toxicity of any ingested NPs. Various techniques based on pre-programmed NPs, including tuning NPs size, shape and surface chemistry (i.e., ligand, charge, hydrophobicity), are being widely explored. Nevertheless, the corona formed in vivo by these methods is always inconsistent with that pre-designed in vitro. This is due to that the pre-designed PC absorbs new species of protein in the biological fluid. Thus, the development of an in vivo corona regulation approach is highly desired to adapt to the complex and dynamic physiological environment. Our work showed that the amount of PC exhibited downtrend during proteolysis process, which meant that the proteolysis treatment inhibited PC association with proteins from the biological fluid and NPs@PC aggregation. As a result, the proteolysis treatment might hold potential to control the PC formation.
- In the introduction, the “low transport efficiency of the substrate into the microbial intracellular enzyme reaction centers” is mentioned. But nothing is said about methods that facilitate the release of proteases through cell membranes.
ANSWER: Thank you for your feedback and question. According to the previous studies,
permeabilizing solvents and detergents, intense submicrosecond electrical pulses, non-thermal atmospheric-pressure plasmas, micrometer-sized graphene oxide, low frequency ultrasound irradiation could enhance cell membrane permeability and facilitate the release of proteases through cell membranes. These methods have been added into the Introdution.
3. The title of the article refers to the “degree of proteolysis”. But in fact, this characteristic is not defined by the authors. Only Figure 1f presents data on the reduction of protein on the surface of nanoparticles. In general, it is customary to use the degree of hydrolysis (DH) as a quantitative characteristic of the progress of proteolysis. The data presented in Figure 1f include information on both processes of hydrolysis of peptide bonds and desorption of peptide fragments.ANSWER: Thank you for your suggestions. Additional experiment of DH has been conducted. DH evaluation of protein corona was also added to out further study on the relationship between protein corona and the cell membrane lipidomics of fermentative bacteria. OPA method with some modifications was used to determine the degree of hydrolysis (DH) of protein corona. 1 ml of 0.1 M sodium tetraborate decahydrate solution containing 0.02 mM SDS, 2 μl of β-mercaptoethanol and 20 of methanol-OPA (1:250, w/v) and 50 μl of proteolytic protein corona solution (0-12 h) or cell-free protease extract from S.cerevisiae were mixed. After 2 min, absorbance at 340 nm was measured, and glycine was used as standard. DH values were calculated as the following formula: NH2ti and NH2t0 were the free amino groups at i and 0 h. NH2ti.CFPE was the free amino groups in CFPE solution at i h. NH2Total was the free amino groups from the whole protein corona.
it was found that DH values of PC started with 0% and gradually increased to 91.6±3.8% during 12 h of CFPE-proteolysis, as show in the following figure (Fig. 1f, red line). The description and the figure have been interpolated into the manuscript. Thank you again for your suggestions.
Lines 236, 245. Should be Figure 1f instead of Figure 1g.
ANSWER: We really appreciate your correction. Fig. 1g had been modified to Fig. 1f in lines 236 and 245.

Reviewer 2 Report
Dear EiC,
Dear Authors,
I read the manuscript with great interest! The paper is correct, results are supported by data, references are adequate and most probably of interest for a quite large group of researchers!
As a material scientists, I would be interested to see also what's happening if US parameters are changes/modified. Can it be exploited in tune the final characteristics/properties of the products? This could further improve the manuscript so, this is why I would like to invite you to make also assessments in this direction!
I would invite the authors to extend, if possible according to my comments!
Best regards,
Reviewer 1
Author Response
What's happening if US parameters are changes/modified. Can it be exploited in tune the final characteristics/properties of the products?
ANSWER: We really appreciate your suggestions. Different species of microorganisms behave various tolerances to US and nanoparticles, and the microstructures of cytoderm and cell membrane in different species of microorganism cell vary so much as to significantly affect the efficiency of US acting on different microorganisms. Hence, a strategic process in US treatment is substantially required to optimized to various species of microorganism. In our previous study, US power of 56.92 w, US duration time of 77 s, nano-Fe3O4 concentration of 341.5 mg/L can maximize the fermentation efficiency of S.cerevisiae, resulting the maximum yield of 2,6-dimethoxy-ρ-benzoquinone. Many thanks for your suggestions.

Round 2
Reviewer 1 Report
The authors took into account all comments. Now the article can be published.